# Liquid Biopsy at the Frontier of Kidney Diseases: Application of Exosomes in Diagnostics and Therapeutics

**DOI:** 10.3390/genes14071367

**Published:** 2023-06-28

**Authors:** Ewud Agborbesong, John Bissler, Xiaogang Li

**Affiliations:** 1Department of Internal Medicine, Mayo Clinic, Rochester, MN 55905, USA; agborbesong.ewud@mayo.edu; 2Department of Biochemistry and Molecular Biology, Mayo Clinic, Rochester, MN 55905, USA; 3Department of Pediatrics, University of Tennessee Health Science Center and Le Bonheur Children’s Hospital, Memphis, TN 38105, USA; jbissler@uthsc.edu; 4Children’s Foundation Research Institute, Le Bonheur Children’s Hospital, Memphis, TN 38105, USA; 5Pediatric Medicine Department, St. Jude Children’s Research Hospital, Memphis, TN 38105, USA

**Keywords:** liquid biopsy, urinary exosomes, biomarkers, therapeutics, kidney diseases

## Abstract

In the era of precision medicine, liquid biopsy techniques, especially the use of urine analysis, represent a paradigm shift in the identification of biomarkers, with considerable implications for clinical practice in the field of nephrology. In kidney diseases, the use of this non-invasive tool to identify specific and sensitive biomarkers other than plasma creatinine and the glomerular filtration rate is becoming crucial for the diagnosis and assessment of a patient’s condition. In recent years, studies have drawn attention to the importance of exosomes for diagnostic and therapeutic purposes in kidney diseases. Exosomes are nano-sized extracellular vesicles with a lipid bilayer structure, composed of a variety of biologically active substances. In the context of kidney diseases, studies have demonstrated that exosomes are valuable carriers of information and are delivery vectors, rendering them appealing candidates as biomarkers and drug delivery vehicles with beneficial therapeutic outcomes for kidney diseases. This review summarizes the applications of exosomes in kidney diseases, emphasizing the current biomarkers of renal diseases identified from urinary exosomes and the therapeutic applications of exosomes with reference to drug delivery and immunomodulation. Finally, we discuss the challenges encountered when using exosomes for therapeutic purposes and how these may affect its clinical applications.

## 1. Introduction

In the last decade, precision medicine has emerged as a powerful clinical strategy for various diseases such as kidney diseases, providing a significant improvement in the evolution and outcome of patient management. To date, plasma creatinine, the glomerular filtration rate, and tissue biopsy have represented the standards for the diagnostics of kidney diseases [1,2]. However, these methods present with several complications, such as being insensitive, providing a single snapshot in time of the disease, and being invasive [3]. To overcome the limitations of conventional biopsy, liquid biopsy has emerged as a non-invasive approach for diagnosing and monitoring patients [4]. Liquid biopsy is the process by which biofluids such as blood and urine are used to assess and diagnose diseases [5]. This examination method permits the measurement of biomarkers such as cell-free DNA (cfDNA) and exosomes [6]. Owing to its non-invasiveness, liquid biopsies have the potential to be developed as a screening tool as they allow for the detection and real-time monitoring of disease progression. Urine is the second most commonly used biofluid for clinical diagnostics. Produced by the kidneys, urine serves to eliminate organic and inorganic waste products from the body and, in so doing, helps to maintain body homeostasis. An advantage of working with urine is the fact that it can be collected in a noninvasive manner. However, urine contents and concentrations may vary as a result of factors such as diet, fluid intake, health status, gender, and time of collection, making the data interpretation of urine analysis for diagnostic purposes complicated, hence the need to identify alternative ways of analyzing urine biopsy.

Aside from the standard wastes found in urine, the serendipitous discovery of extracellular vesicles (EVs)/exosomes opened up new possibilities for the use of urine in clinical diagnostics since the EVs/exosomes contain molecules not previously detected in urine [7]. Initially identified in the 1980s and believed to function in eliminating cellular wastes resulting from homeostasis or damaged cells [8], it is now known that exosomes are more than just waste carriers, with the capacity to act as signaling vehicles, delivering complex cargo of nucleic acids [9,10], proteins [11], and lipids [12,13,14] to target cells, eliciting functional responses and promoting phenotypic changes that ultimately reprogram the recipient cells. In addition, the initial characterization of urinary EVs/exosomes identified proteins that were typical of epithelial and urothelial cells of the nephron [15]. As such, exosomes offer prognostic information in a wide range of diseases, such as inflammatory diseases [16,17], tumors [18], and renal diseases [19,20], and play a major role as diagnostic and therapeutic tools by affecting diverse cellular processes such as signal transduction [21], antigen presentation [22], and immune regulation [23,24]. Exosomes generated considerable interest for clinical application as diagnostic and therapeutic tools for several reasons. Their presence in diverse types of biofluids gives them the potential to be identified as biomarkers. In addition, their small size and membrane composition permit them to cross major biological membranes including the blood–brain barrier [25,26,27,28]. Furthermore, their biocompatibility, which reduces immunogenicity, and their lipid bi-layer structure, which protects cargo from degradation, makes them attractive as therapeutic vectors [29].

In recent years, updated technical means in liquid biopsy have made it available for wide application in the clinical diagnosis of a range of diseases. Particularly, isolates from urinary exosomes led to the discovery of biomarkers in diseases such as prostate cancer [30] and bladder cancer [31,32]. Such promising studies, coupled with the fact that urinary EVs/exosomes were considered to originate from cells of the urogenital tract, sparked an exponential growth in exosome research with the goal of discovering EVs/exosome biomarkers for urogenital-tract-related pathologies, providing new possibilities for diagnostic and therapeutic applications of EVs/exosomes in kidney diseases. The diverse cell origins and molecular compositions of urinary exosomes, however, pose analytical difficulties. Therefore, standardizing the methodologies for urine sample collection and EV/exosome isolation would go a long way in the discovery of biomarkers and provide insights that would guide sound clinical decisions. With this viewpoint, this review summarizes advances made in exosome research as it pertains to kidney-related diseases, including acute kidney injury, chronic kidney disease, polycystic kidney disease, and renal cell carcinoma. We give a brief overview on the characterization and isolation of extracellular vesicles. In addition, we discuss urine sample handling and processing and provide recommendations to improve the experimental reproducibility. We further summarize updates on the use of exosomes as a diagnostic and therapeutic tool for targeted kidney diseases, with emphasis on their roles in the identification of biomarkers, drug delivery, and immunomodulation. Finally, we address the current practical challenges encountered with the use and clinical translation of exosomes in kidney diseases. Addressing these issues may provide new mechanistic insights and enable a more sophisticated translation of the use of urinary exosomes as novel biomarkers and therapeutic intervention strategies in kidney diseases.

## 2. Biogenesis, Composition, Characterization, and Function of Exosomes

### 2.1. Heterogeneity of Extracellular Vesicles

In the early 1980s, the concept that EVs are formed and released through the budding of the plasma membrane was replaced by the concept that intercellular vesicles mediated by multivesicular bodies (MVBs) are formed and released into the extracellular medium [33,34]. Several years later, the term “exosomes” was coined to refer to these EVs (30–100 nm in diameter) of endosomal origin [8]. Today, based on transmission and immuno-electron microscopy and biochemical methods [35,36], EVs may be broadly divided into two main categories: exosomes and microvesicles. However, the classification of EVs is continuously evolving and newly discovered organelles including migrasomes, generated from migrating cells [37,38], and apoptotic bodies, generated by apoptotic cells [39], are now classified as EVs (Figure 1). It is worth noting that, in addition to apoptotic bodies, dying/apoptotic cells can also release microvesicles and exosomes [40].

EVs in urine originate from different parts of the urogenital tract, including the kidneys, bladder, vaginal tract, and prostate. Urinary EVs may also originate from residing immune cells and bacteria, and also from circulation [41], though it is not clear how these EVs enter the urine. EVs possess unique characteristics, such as their size, morphology, density, presence of marker proteins, and biogenesis (Table 1). For example, exosomes and microvesicles differ in the mechanism of their biogenesis, with exosomes being generated by the inward budding of the endosomal membrane as intraluminal vesicles (ILVs) whereas microvesicles originate by an outward budding at the plasma membrane [42] (Figure 1), and more recently identified from the primary cilium [43,44]. In general, owing to their cellular origin and composition, EVs have been found to play important roles in a variety of cellular processes, such as immune regulation, development, regeneration, and reproduction [45,46,47,48,49], and, as such, are associated to the pathogenesis of a variety of diseases [50]. However, exosomes have received more attention than the other EVs and will be the focus of this review.

### 2.2. Exosome Biogenesis

The biogenesis of exosomes is strictly regulated such that every step of EV formation is directed by specific components (Figure 2). Exosome biogenesis starts with the formation of an early endosome that matures into a late endosome [61,62]. During endosomal maturation, ILVs are formed via endosomal sorting complex required for transport (ESCRT)-dependent and independent machinery [36,63]. The late ILV-containing endosomes are referred to as multivesicular endosomes (MVEs) or multivesicular bodies (MVBs). The ILVs that eventually become exosomes harbor specific proteins and nucleic acids that are stochastically acquired and/or sorted cytoplasmic and membrane-bound contents originating from processing steps within the donor cell [64,65,66].

Generally, MVBs will fuse with lysosomes to degrade or recycle their contents. The MVBs that do not undergo lysosomal degradation are secreted from the cell via Rab guanosine triphosphatases (Rab-GTPases) [62,67], enhanced by other factors, such as pH levels [68] and calcium concentration [69] within and/or outside the donor cell. The secretion of MVBs is accompanied by the release of exosomes (Figure 2) into the extracellular environment [70], where they may modulate the recipient cell’s activity in an endocrine or paracrine manner [71]. The orientation and distribution of lipids in the bi-layered membrane that result from the invagination of the plasma membrane during the formation of the endosome and subsequent ILVs are crucial to the exosomes’ ability to efficiently mediate cell–cell communications with recipient cells [72]. The lipid bilayer appears to be asymmetrical in the exosome membrane, with phosphatidylserine species typically found in the inner leaflet and sphingomyelin species in the outer leaflet [73]. However, phosphatidylserine is reported to be externalized in apoptotic and malignant cells, acting as an “eat me” signal for macrophages in the immune system [74]. The ordered distribution of lipids in the exosomal membrane, on the other hand, is suggested to be responsible for interactions during exosome formation, release, and delivery to recipient cells. In this regard, studies have revealed that exosomes modify their lipid and metabolite composition depending on the conditions under which they are produced [75,76].

### 2.3. Exosome Composition

The cellular origin and size of exosomes determine their composition, which may dynamically change as a result of modifications that occur in their parent cell. Exosomes are limited by a bilayered lipid membrane that encapsulates cargo molecules in an inner aqueous core. The lipid composition of exosomes is derived from different sources, which shows that the exosomal lipid membrane is a highly ordered structure enriched in varying lipid classes such as cholesterol and glycosphingolipids [14,77]. This enrichment in lipids is proposed to increase the exosomes’ stability in extracellular environments. The ordered distribution of lipids in the exosomal membrane is believed to play a vital role in the formation of interactions required for exosome biogenesis, release, and delivery/signal transduction to recipient cells [14]. The lipid distribution into exosomes is a dynamic process that responds to several factors and may be modified under different conditions. In one study, for instance, significant variations in the lipid composition of reticulocyte-derived exosomes were observed in response to the physiological changes in the cell during the maturation to erythrocytes, which suggests that lipid sorting for exosome biogenesis could adapt to the cell requirements [78]. In another study, it was demonstrated that prostate cancer PC-3 cells cultured with the ether lipid precursor hexadecylglycerol secreted exosomes enriched in ether lipids, which suggests that external stimuli have the capability to impact the lipid composition of exosomes [75].

The cargo composition of exosomes is unique and complex, differing from one donor cell type to another, the physiological stage of the donor cell, and the fate and function of the exosome [14,78]. Primarily, the cargo consists of peptides, small proteins, and nucleic acids (DNA and RNAs) [11] (Figure 3), all of which are used by the donor cell to transmit signals to other cell populations, coordinate biological functions, and maintain homeostasis. The most frequently identified proteins in exosomes are tetraspanins (e.g., CD9, CD63, and CD81), MVB biogenesis proteins (e.g., Alix and TSG101), membrane transporters and fusion proteins (e.g., GTPases, annexins, and flotillin), heat shock proteins (e.g., HSC70), and lipid-related proteins and phospholipases [11,79]. Several proteins are recognized as specific exosomal markers, among which the tetraspanins CD63 and CD81 are most commonly used [80]. Currently, the exosome content database, Exocarta (http://www.exocarta.org accessed on 16 March 2023), hosts 41,860 protein, >7540 RNA, and 1116 lipid-cataloged entries from 10 distinct species. From these entries, 9769 proteins, 3408 mRNAs, 2838 miRNAs, and 194 lipids have been identified in exosomes from multiple organisms [65,81].

The composition of the exosome provides information about the state of the cell of origin and offers clues regarding its possible effects on the recipient cells [82]. In the kidney, for example, the presence of integrins has been identified on various cell types, including fibroblasts [83] and tubular epithelial cells (TECs) [84,85]. Under physiological conditions, TECs have been found to express αv and β1 integrins. However, under pathological conditions such as kidney injury, they also express β6 in addition to αv and β1 integrins [86]. Similarly, fibroblast cells are known to express α1, α4, α5, and β1 integrins under physiological conditions. However, under pathological conditions associated with fibrosis, they express αv, α5, and β1 integrins [87]. Integrin αv expressed by fibroblast cells has been reported to bind TGF-β, resulting in the stimulation of fibrosis [88], while an increased expression of β1 integrin is associated with epithelial cell polarization [89]. Therefore, one could speculate that, under pathological conditions, exosomes containing αv integrin would stimulate fibrosis in target cells while exosomes containing β1 integrin would enhance TEC polarization and detachment from the basal membrane. It is important to note, however, that the effects of exosomes on target cells may differ due to the varying expressions of cell surface receptors found on different target cells, creating a functional heterogeneity [90]. Likewise, exosome heterogeneity may also arise from differences in the tissue and organ of origin of the exosome.

### 2.4. Urine Handling for Exosome Research

In urinary exosome studies, the pre-analytical handling of the urine sample is a major source of data variability and can limit reproducibility as preservation and storage methods may impact the outcome [91]. Thus far, there are very limited studies addressing urine sample collection, processing, and storage for exosome research. Nevertheless, pre-analytical variables such as collection (use of preservatives), processing (centrifugation), storage (short-term storage at 4 °C or immediate freezing at −80 °C), and urine handling after thawing (slow or rapid thawing) can alter the results and limit reproducibility. Therefore, consistency in handling urine samples within a study and across research teams is of utmost importance. Due to study specifications, individual research groups have been known to collect, process, and store urine samples differently, making it difficult to come up with a universal standard for urine sample handling in exosome research. In a recent study, however, the authors compared the current methods used for urine collection and storage for urinary EV utilization. They concluded that the addition of protease inhibitors and preservatives such as sodium azide, long-term storage at −80 °C, and extensive vortexing after thawing provides the best quality for urinary EVs [92].

### 2.5. Exosome Isolation and Characterization

The isolation of pure exosomes is critical to understanding their mode of action and for their potential clinical applications. However, due to their small size and heterogeneity in the exosome population, the isolation of exosomes is challenging. In urine, for example, exosomes originate from different parts of the urogenital tract, including the kidneys, bladder, vaginal tract, and prostate, as well as residing immune cells and bacteria, and also from the circulation [43]. Moreover, within an organ such as the kidney, exosomes can be released from different cell types, such as epithelial and stromal cells. Nevertheless, various techniques have been adopted to facilitate the isolation and characterization of exosomes. Isolation techniques include differential ultracentrifugation [93], size exclusion chromatography [94], immune affinity capture [95,96], exosome precipitation [97], microfluidic-based isolation [98], polymer precipitation [99], and commercially available kits (Table 2). Currently, centrifugation-based techniques are widely considered to be the gold standard for exosome isolation. However, the most essential method for size-based isolation is ultrafiltration [100]. Size exclusion chromatography enables size-based separation on a single column and ensures exosome elution before other soluble components such as proteins [101]. Together with ultrafiltration, size exclusion chromatography techniques are reported to achieve exosome preparations of high purity [102]. It is obvious that each of these isolation techniques has its pros and cons.

Once isolated, exosomes are characterized based on their size, morphology, density, and the presence of marker proteins. However, due to the differences in isolation techniques, heterogeneity of the exosome population, and the difficulty in completely separating cargo profiles, the characterization of exosomes can be particularly challenging. In general, the method used for characterizing exosomes depends on their specific attributes. For size characterization, for example, available methods include nanoparticle tracking analysis (NTA) [118,119] and dynamic light scattering (DLS) [120]. For the characterization of exosome morphology, electron microscopy is the standard method. The morphology observed by scanning electron microscopy (SEM), however, contradicts that of transmission electron microscopy (TEM). While SEM images depicts round-shaped exosomes, TEM images show that exosomes are cup-shaped [118,120]. This difference may be as a result of the type of information provided by the techniques. While SEM provides a three-dimensional image of the surface of the sample, TEM provides a two-dimensional projection image of the inner surface of the sample [121]. It is also important to note that TEM requires very thin sections that are generally less than 150 nm and, in some instances, even less than 30 nm for higher-resolution images [121]. Therefore, sample preparation is quite complex and tedious and may affect exosome properties. For proteomics and biological characterization, mass spectrometry and Western blotting have been widely used.

### 2.6. Advantages of Exosomes in Liquid Biopsy

Due to their unique features, exosomes have become promising analytes in the field of liquid biopsy. Among the different components such as cfDNA that can be analyzed in liquid biopsy, exosomes are particularly promising because of several reasons. First and foremost, exosomes exist in almost all biofluids and are known to be highly stable owing to their lipid bilayer. This stability can have an effect on the sample storage and transportation, which impacts the clinical applicability of exosomes [122]. Second, exosomes contain the biological information from their parent cells, opening up a valuable avenue for future genetic studies/screening and human disease diagnosis and monitoring. Third, exosomes are relatively easy to identify since they express biological contents such as proteins and lipids that can be used as markers to differentiate them from other extracellular vesicles [123]. Moreover, the specificity of the surface proteins from their parent cells may help to predict the origin of the exosome and provide information on organ-specific disease [124]. Fourth, compared to other liquid biopsies such as circulating tumor cells (CTCs), exosomes are relatively easier to obtain from other biofluids [125]. In addition, it has been reported that a considerable amount of human plasma cfDNA are located in exosomes [126] and that the mitochondrial DNA copy number is higher in exosomes than in plasma in some disease [127]. Fifth, reports indicate that the frequency of detecting mutations in exosomal DNA is higher than that in cfDNA and that exosomal DNA has greater prognostic value compared to cfDNA [128,129]. Finally, exosomes are secreted by living cells, unlike cfDNA, which is secreted during cell death (apoptosis and necrosis). As such, exosomes contain biological information from their parent cells, rendering them more representative and clinically applicable. Undoubtedly, exosomes play an important role in various physiological and pathological processes, and there is evidence to show that exosomes are potential tools for clinical application, including liquid biopsy. Whether there is a difference between urine and blood exosomes as diagnostic biomarkers in kidney disease remains unclear. It has, however, been reported that exosome concentration in urine is substantially lower compared to serum [130]. In addition, exosomes from any biofluid are a representation of the cells in that given microenvironment. Therefore, according to human anatomy and pathophysiology, urinary system diseases would benefit more from urine than blood, suggesting that exosomes from urine would be better diagnostic biomarkers in kidney disease.

## 3. Exosome-Mediated Mode of Communication

In general, neighboring cells communicate with one another through direct cell–cell contact, such as gap junctions and cell surface protein/protein interactions, while distant cells communicate with one another through secreted soluble factors, such as hormones and cytokines [131,132]. In recent years, exosomes, with their cell-specific cargo, have been recognized as a novel mechanism of intercellular communication [133,134]. Exosomes can mediate cell–cell communication locally and systemically [135]. Once released into the extracellular microenvironment, exosomes can reach the recipient cells and promote phenotypic changes or trigger responses [132,135]. The success of exosomal communication is highly dependent on the effective mechanism of cargo delivery, which may be achieved via receptor–ligand interactions, direct fusion with the plasma membrane of the recipient cell, or via internalization [136] (Figure 4). As such, the exosomes impact recipient cells by the direct stimulation of surface-bound ligands, the transfer of activated receptors to recipient cells, and by epigenetic reprogramming through the delivery of functional epigenetic proteins and miRNAs [137,138,139,140].

Receptor–ligand interactions are a common route for the mediation of immunomodulatory functions [141]. Transmembrane ligands on the exosome surface bind to receptors on the surface of the recipient cells and, depending on the nature of the ligand and receptor, specific downstream signaling cascades are subsequently activated or inhibited [142]. Exosomes also fuse directly to the recipient cell and release their contents into the cytosol [143]. Similar to cell membrane fusion, the lipid bilayer of the exosome partially fuses with that of the recipient cell with the help of Rab GTPases and SNARE family proteins, and eventually expands to form a consistent structure [144]. Exosomes can also undergo internalization by the recipient cell. The process of internalization may occur via various pathways, including clathrin-coated endocytosis, caveolin-mediated endocytosis, lipid-raft-mediated endocytosis, and phagocytosis [143,145,146,147]. However, the fate of the exosomes post-internalization is following the endosomal pathway and eventually fusing with the lysosome for degradation [136,148], or bypassing degradation and being directed to other cellular locations where they mediate functional changes [136].

The role of exosomes in cell–cell communication has been extensively studied and found to have multiple physiological and pathophysiological functions. It is plausible to suggest that these functions are associated with the route of exosome–recipient cell communication. It is also plausible that the mode of communication may be determined by the donor cell type, the exosome membrane composition, and/or the cargoes.

## 4. Exosomes as Biomarkers for Kidney Diseases

The most commonly used markers for kidney diseases include the estimated glomerular filtration rate (eGFR), blood urea and proteinuria, albuminuria, and serum creatinine. However, these markers are unable to reflect functional changes in the kidney at early disease stages. Identifying novel biomarkers may overcome this limitation and provide a clearer understanding of kidney pathophysiology. In this light, urine contains a considerable number of exosomes that may reflect changes in different cellular compartments. As such, urinary exosomes have been suggested as a promising liquid biopsy for kidney diseases [149]. The composition of urinary exosomes secreted from different segments of the nephron and their relevance to the kidney physiology and pathology of kidney diseases have been investigated. Proteomic analysis of exosomes secreted under physiological and pathological conditions revealed significant changes in protein expression. Cargos released in pathological conditions, such as cancer and inflammation, contain specific constitutive components, such as transmembrane proteins and nucleic acids, which can act as biomarkers for clinical diagnosis, staging disease severity, or assessing therapeutic response [150,151]. Studies from the past few years have identified exosome proteins as biomarkers for different kidney diseases (Table 3). The roles of EVs in renal physiology have been investigated both in vivo and in vitro. EVs are produced and secreted by many cell types in the kidney and proteomic analysis of urinary EVs has demonstrated that the contents of these EVs arise from all segments of the kidney, including proximal tubules, the distal tubule, and the collecting duct [15]. To date, urinary exosomes have been proposed as a promising source of non-invasive biomarkers for the diagnosis and prognosis of various kidney diseases. Urinary exosomes were suggested since they act as indicators of renal function as exosomes secreted by TECs into the urine may vary depending on the physiological and pathological state of the kidney [152,153]. Furthermore, based on the origin, the composition of urinary exosomes may serve as biomarkers for specific kidney diseases [154]. In this section, we discuss the identification of urinary exosomal biomarkers in several common types of kidney diseases, including acute kidney injury, chronic kidney disease, polycystic kidney disease, and renal cell carcinoma.

### 4.1. Acute Kidney Injury (AKI)

Acute kidney injury (AKI) is a common clinical condition associated with the risk of developing chronic kidney disease (CKD) and end-stage kidney disease (ESKD). Sepsis is the leading cause of AKI in the intensive care unit (ICU) and accounts for 45–70% of all AKI cases [182]. Traditionally, the diagnosis of acute kidney injury (AKI) relies on the serum creatinine and urine output. However, this diagnostic tool is believed to be less sensitive and specific. Since exosomes are secreted by live cells, the detection of urinary exosomes and their contents are explored as potential specific biomarkers for AKI (Table 3). In renal ischemia–reperfusion (I/R) injury, for example, the decreased expression level of aquaporin-1 (AQP1), believed to be controlled by urinary exosomes, is suggested to be a novel urinary biomarker for renal (I/R) injury [155]. In cisplatin-induced AKI, the organic anion transporter 5 (Oat5) was identified as a biomarker for AKI. It was demonstrated that the urinary excretion of exosomal was notably increased, but, when renal injury was ameliorated by the administration of N-acetylcysteine, an Oat5 increase was undetected [156]. Similarly, a significant increase in urine exosome fetuin-A was detected in cisplatin-induced AKI prior to any evidence of morphological injury, suggesting the potential for fetuin-A to serve as a biomarker in AKI patients [183]. In acute tubular necrosis, increased urinary exosome levels of Na^+^/H^+^ exchange type-3 were observed and identified as a potential biomarker for AKI [158]. Additionally, urinary exosomal levels of the activating transcription factor 3 (ATF3) were reportedly elevated in AKI patients, even before serum creatinine levels, suggesting that ATF3 as a biomarker may be used for the early diagnosis of AKI [157].

In a sepsis-induced AKI rat model, the microRNAs miR-181a-5p and miR-23b-3p were found to be differentially expressed in circulating extracellular vesicles earlier than creatinine elevation. The expression of these miRNAs may potentially serve as an important tool for the early identification of sepsis-induced AKI and for discriminating sepsis-induced AKI from other causes of AKI [184]. In AKI patients with cirrhosis, the upregulation of maltase–glucoamylase (MGAM), a renal brush border disaccharidase, in urinary exosomes has also been suggested as a potential biomarker, which may differentiate the type of kidney injury in cirrhosis. However, its clinical relevance needs to be further validated [185]. Several other urinary biomarkers have been identified as indicators for the prediction and diagnosis of AKI, including liver fatty-acid-binding protein (L-FABP) [186], neutrophil gelatinase-associated lipocalin (NGAL) [187,188], kidney injury molecule-1 (KIM-1) [189], N-acetyl-β-D-glucosaminidase (NAG) [190], tissue inhibitor metalloproteinase-2 (TIMP-2), insulin growth factor binding protein-7 (IGFBP-7) [191,192], and urinary thioredoxin [193]. Though these biomarkers have been identified, their availability for clinical care has proven slow due to accuracy, the availability of testing platforms, variability in assay techniques, and the cost. It is important to note, however, that IGFBP-7 [194] and thioredoxin [195] are clinically available.

### 4.2. Chronic Kidney Disease (CKD)

Chronic kidney disease (CKD) is characterized by renal dysfunction, usually diagnosed by the estimated glomerular filtration rate (eGFR) and albuminuria [196]. In recent years, exosomes have been extensively investigated as diagnostic tools in CKD, particularly in renal fibrosis and diabetic nephropathy (Table 3). In renal fibrosis, for example, several studies have identified urinary exosomal micro RNAs in CKD pathology. Urinary exosomal miR-29c [160], miR-181a [162], and miR-200b [163] were reportedly decreased in CKD patients, correlating with the degree of renal fibrosis. By contrast, increased levels of secreting transglutaminase-2 were identified in a unilateral ureteral obstruction (UUO) mouse model [164]. In addition, urinary exosomal miR-21 was increased in CKD patients and inversely correlated with eGFR [197]. Urinary exosomal ceruloplasmin was also found to be increased in CKD rats and patients. Furthermore, immune-reactive ceruloplasmin localized in tubules and collecting ducts of biopsies of CKD patients, while its increased levels were detected in the animals prior to the onset of proteinuria [198]. In more recent years, studies have reported the identification of circular RNA (circRNA) as biomarkers for CKD diseases. The expression level of hsa_circ_0008925 was increased in urinary exosomes from glomerular diseases [165]. In another study, human circRNA analysis reported an increased expression level of hsa_circ_0036649 in urinary exosomes in glomerular diseases [166]. However, its potential role as a biomarker in CKD remains to be further validated.

In addition to renal fibrosis, several proteins and miRNAs have also been identified as biomarkers in diabetic nephropathy. In a proteomic study using label-free comparative techniques to analyze urinary exosomes, it was demonstrated that the urinary exosomal bikunin precursor and histone-lysine N-methyltransferase were increased, whereas voltage-dependent anion-selective channel protein 1 (VDAC1) was decreased in diabetic nephropathy [167]. Additionally, increased levels of AQP-2 and AQP-5 were detected in urinary exosomes derived from diabetic nephropathy patients [168]. These data suggest that these proteins may serve as non-invasive biomarkers for the diagnosis of diabetic nephropathy. Subsequent microarray analysis identified a number of miRNAs as biomarkers for diabetic nephropathy. Increased levels of microRNAs such as miR-371b-5p, miR-320c, miR-572, miR-1234-5p, miR-6068, miR-6133, miR-4270, miR-4739, and miR-638, among others [169], are elevated in urinary exosomes derived from type 2 diabetic nephropathy patients. miR-30d-5p and miR-30e-5p, however, were decreased in type 2 diabetic nephropathy patients [169]. In another study, urinary exosomal levels of several miRNAs, including miR-15b, miR-30a, miR-34a, miR-133b, miR-342, and miR-636, were increased in type 2 diabetic nephropathy patients [170,171]. Additionally, the urinary exosomal level of let-7c-5p was found to be increased, while miR-15b-5p and miR-29c-5p were decreased in type 2 diabetic nephropathy patients [172]. These findings underscore the importance of urinary exosomes and, more particularly, the role of micro RNAs as the source of non-invasive biomarkers for the diagnosis of CKD.

### 4.3. Polycystic Kidney Disease (PKD)

Polycystic kidney disease (PKD) is the most common inherited kidney disease and is predominantly caused by a mutation in the genes encoding for polycystin-1 (PC-1), polycystin-2 (PC-2), and fibrocystin/polyductin (FCP), all of which are involved in primary cilia structure and function [199]. Proteomic analysis of PKD urinary exosome-like vesicles detected gene products involved in PKD, suggesting that exosomes may play a significant role in primary cilia biology and may serve as a biomarker for PKD. Subsequent studies demonstrated that ADPKD patients with PKD1 gene mutation had decreased levels of polycystin-1 and polycystin-2 but increased level of transmembrane protein 2 in urinary exosomes [15,173,174]. In addition, cystin, the product of the mouse cpk locus, and ADP-ribosylation factor-like 6, the product of the human Bardet–Biedl syndrome gene (BBS3), were found to be abnormally expressed in urinary exosome-like vesicles of patients with PKD [173]. Further proteomic analysis demonstrated increased levels of complement C3 and C9 in urinary extracellular vesicles derived from ADPKD patients with or without progressive CKD and increased levels of envoplakin, periplakin, and villin-1 only in exosomes from ADPKD patients with progressive CKD [175]. This suggests that envoplakin, periplakin, and villin-1 may be used as biomarkers to differentiate between autosomal dominant polycystic kidney disease (ADPKD) patients with or without CKD.

In another study, the activator of G protein signaling 3 (AGS3), involved in the regulation of polycystin ion channel activity, adenylyl cyclase activity, mitotic spindle orientation, and programmed cell death [200,201], was demonstrated to be significantly increased in urinary exosomes from PCK rats and patients [176]. Additionally, AGS3 was suggested to be a beneficial repair protein in tubular epithelial cells [202], which could facilitate the trafficking of proteins to the plasma membrane. This suggests that AGS3 could play a potential role in the trafficking of proteins, enabling exosomal communication between tubular epithelial cells in PKD. In a more recent study, increased urinary exosomal levels of prominin 1 (CD133), the cellular repressor of E1A-stimulated genes 1 (CREG1), and cadherin 4, indicating morphological and proliferation aberrations, were found to be elevated in the exosomes of patients with ADPKD [177]. Notably, increased urinary exosomal CD133 was identified as a potential biomarker that could be used to distinguish medullary sponge kidney disease patients from ADPKD patients [177].

These studies demonstrate that, in PKD, exosome subpopulations with unique compositions trigger diverse biological effects that affect neighboring cells. As a matter of fact, this was reported in a recent study, where the authors demonstrated that cystic-cell-derived extracellular vesicles and urinary exosomes derived from ADPKD patients promoted cyst growth in *Pkd1* mutant kidneys and in 3D cultures [19]. Together, these studies draw attention to the key role of exosomes and demonstrate that urinary exosome contents may: (i) serve as biomarkers for PKD, (ii) have the potential to provide a non-invasive method to monitor the progression of PKD, and (iii) provide insight into the biology of tubular epithelial cell function during PKD disease progression.

### 4.4. Renal Cell Carcinoma (RCC)

Renal cell carcinoma (RCC) is one of the most common cancers, with an incidence of approximately 3.7% of all new cancer cases. As such, it is important to be able to predict the symptoms of the disease before the appearance of symptoms. Comparative proteomics analysis of urinary exosomes from RCC patients and healthy controls identified several urinary exosomal protein markers, amongst which 10 were validated by Western blot analysis, including matrix metalloproteinase-9 (MMP-9), ceruloplasmin (CP), podocalyxin (PODXL), Dickkopf-related protein 4 (DKK 4), carbonic anhydrase IX (CAIX), aquaporin-1 (AQP-1), neprilysin (CD10), dipeptidase 1 (DPEP1), extracellular matrix metalloproteinase inducer (EMMPRIN), and syntenin-1 [178]. Additionally, altered levels of microRNAs have been identified as biomarkers in RCC. In clear cell RCC (ccRCC) patients, for example, increased levels of miR-150-5p and decreased levels of miR-126-3p have been reported. In this same study, it was reported that a combination of urinary miR-126-3p and miR-449a could be used to distinguish between healthy individuals and ccRCC patients with high sensitivity [179]. In another study, urinary exosomal miR-30c-5p and miR-204-5p were identified as potential diagnostic biomarkers for early-stage ccRCC [180,181].

In summary, exosomal contents such as proteins and miRNAs in urine and plasma make it possible to predict the onset and progression of disease. However, their translation to clinical practice faces some challenges. First of all, the isolation methods need to be optimized and standardized to improve reproducibility. Second, the sensitivity and specificity of exosomal components need to be further analyzed and confirmed. Finally, to ensure the accuracy and reliability of the results, more patients need to be enrolled in the clinical trials.

## 5. Exosomes and Therapeutic Potential

One of the features that make exosomes useful in diagnostics also makes them useful therapeutically. Exosomal surface markers can be used to target specific cell types. Whether for drug delivery or gene therapy, exosomes engineered for a chosen cell type can be packaged with desired components. Stem cell therapy represents a promising new avenue in the treatment of kidney diseases. Several types of stem cells, including induced pluripotent stem cells (iPSCs) and mesenchymal stem cells (MSCs), have been shown to attenuate kidney dysfunction [203]. In recent years, human urine-derived stem cells (USCs) were introduced as a novel promising candidate in the treatment of kidney diseases. Patient-derived USCs are isolated from urine and studies have found that these USCs could be used as a tool to predict the outcome of the kidney disease [204]. USCs are an attractive cell source for a variety of therapies because they are easily accessible, can be consistently produced by the patient, and are relatively free of ethical concerns. USCs exhibit many characteristics of MSCs, and studies have shown that MSCs play an important role in both acute and chronic kidney diseases. Furthermore, treatments with MSCs have entered clinical trials [205]. The following section focuses on the progress of research on the use of exosomes for drug delivery and immunosuppression in kidney diseases, with particular emphasis on MSCs.

### 5.1. Exosome Mediated Drug Delivery

Exosomes are acknowledged to function in intercellular communication, triggering physiological responses. Once internalized, the exosome contents can be directly released into the cytosol or fuse with the endosomal membrane, evoking an effect on the recipient cell [143]. This inherent ability of exosomes to transfer biochemical materials between cells highlights their potential as drug delivery vehicles. Several studies have investigated the functionality of exosomes in the delivery of materials, including miRNAs [206], siRNAs [207,208,209], and shRNA [210]. Furthermore, their performance as therapeutic vehicles for the delivery of the anti-inflammatory agent curcumin [211,212] and anticancer agents, including paclitaxel [213] and doxorubicin [214], have been investigated. Compared to other delivery systems such as viruses, liposomes, and nanoparticulate systems, known to have undesirable properties, such as immune activation as foreign particles and toxicity, exosomes have been found to increase the loading efficacy of doxorubicin and decrease the adverse effects on major organs, particularly the heart [214,215]. This suggests that doxorubicin delivery via exosomes may alleviate adverse effects, making them a viable vehicle for the chemotherapeutic drug. Furthermore, this suggests that drug delivery via exosomes may be a viable therapeutic tool for the treatment of kidney diseases.

#### Drug Loading into Exosomes

Considerable advancements have been made in the development and validation of exosome-based drug delivery systems in recent years. Several approaches have been utilized for the loading of exosomes with therapeutic cargoes. The most used approaches may fall under one of the following categories: (i) the ex vitro loading of naïve exosomes isolated from parent cells; (ii) drug loading into parent cells; and (iii) genetically modifying parent cells with DNA encoding therapeutically active compounds (Table 4). In both categories (ii) and (iii), where the parent cells are altered, the drug and therapeutically active compounds are released in the exosomes.

i.Ex vitro loading of exosomes isolated from parent cells: this is a passive approach of loading exosomes with small molecules (typically lipophilic), nucleic acids, and proteins by co-incubating these compounds with the exosome. The simple incubation of small molecules such as doxorubicin and paclitaxel may easily penetrate the exosome’s membrane. Nucleic acids and proteins, however, require reformation and reshaping techniques such as sonication, electroporation, and elevating incubation temperatures to achieve high loading efficiencies. The ex vitro loading of exosomes approach seems to be the most practical since exosomes obtained from several isolations may be pooled and then loaded with therapeutic cargo.ii.Drug loading into parent cells: in this approach, an exogenous compound is loaded into the parent cell and subsequently released in exosomes into the conditioned medium. For example, it has been demonstrated that mesenchymal stem cells (MSCs) treated with paclitaxel secrete exosomes that, in turn, contain paclitaxel [216]. It should be noted, however, that mammalian cells release low quantities of exosomes, and the purification is usually difficult and with a low yield. To circumvent this, bioinspired exosome-mimetic nanovesicles were developed to target and deliver chemotherapeutic drugs [217]. In this study, the nanovesicles were produced by the breaking down of monocytes or macrophages using a serial extrusion technique. Compared to exosomes, these cell-derived nanovesicles have a 100-fold higher production yield. Furthermore, they have a natural targeting ability since they maintain the topology of the plasma membrane proteins of the originating cell [217]. The engineering of parental cells via liposomes has been demonstrated to selectively deliver hydrophobic compounds to the plasma membrane of cancer cells [218]. In this study, the authors introduced synthetic membrane fusogenic liposomes loaded with chemotherapeutic drugs into parent cells. These liposomes were incorporated into the membranes of vesicles/exosomes and then transferred to neighboring cells [218].iii.Genetically modified parent cells: in this approach, the genetic material of the parent cell is modified to express, delete, or re-direct the localization of a gene product, and these changes are reflected in the secreted exosomes. For example, a drug delivery system was developed where macrophages were transfected with plasmid DNA encoding different therapeutic proteins for the treatment of neurodegenerative disorders [219]. In another system, adeno-associated virus (AAV) capsids were incorporated into extracellular microvesicles termed “vexosomes” (microvesicle-associated vectors). In this study, the authors found that, during the production of AAV vectors, a fraction of released vectors were associated with vexosomes, which enhanced gene transfer in cultured cells compared to conventionally purified AAV [220].

**Table 4 genes-14-01367-t004:** Methods of exosomal bioengineering and their advantages and disadvantages.

Loading Approach		Functions	Advantages	Disadvantages	References
Passive loading	Simple incubation of exosomes with drugs.	Incorporation of lipophilic drugs.	- Easy to perform. Does not affect exosome integrity.	Low loading efficiency.	[211,216]
	Simple incubation of parent cell with drugs.	Incorporation of lipophilic drugs.	- Easy to perform. - Does not affect exosome integrity.	- Low loading efficiency. - May cause cytotoxicity to parent cell.	[216,221]
Active loading	Sonication.	Incorporation of drugs, proteins, and peptides.	High loading efficiency.	Compromises exosome membrane integrity.	[222]
	Electroporation.	Incorporation of drugs and large molecules such as nucleic acids and peptides.	Loading efficiency is high.	May cause RNA aggregation and exosome instability	[223,224]
	Extrusion.	Incorporation of drugs.	High loading efficiency.	High loading efficiency.	[222,225]
	Freeze–thaw.	Incorporation of proteins and peptides.	Medium loading efficiency.	- May inactivate cargo. - Aggregates may be formed.	[222]
	Use of membrane permeabilizers (surfactants).	Incorporation of proteins and peptides.	High loading efficiency.	- Compromises exosome integrity. - May inactivate cargo.	[222,225]
	Transfection.	Incorporation of nucleic acids, proteins, and peptides.	High loading efficiency and molecular stability.	Toxicity of transfection agents.	[222,226]

Exosome-based drug formulations may be applied to a wide variety of disorders. However, the choice of loading approach may be dictated by the type of cargo to be loaded, the conditions appropriate for the loading of any given type of cargo, and the site of the disease. Though each approach has its advantages and limitations as summarized in the table above (Table 4), a series of them have been successfully used for exosome-mediated drug delivery.

### 5.2. Exosome-Mediated Drug Delivery in Kidney Diseases

For kidney-related diseases, identifying a prime target for potential exosome-based therapy could have significant implications for therapeutic purposes being that the kidney is made up of several cell types with distinct functions. As such, understanding exosome production, the molecules that they carry, and their interactions within the kidney could have far-reaching implications for therapeutic purposes of kidney diseases. Studies have demonstrated that the purification and use of exosomes from particular cells could be a promising approach for the therapeutic purposes of kidney diseases. An example of this is the isolation and use of exosomes derived from MSCs. MSCs are commonly used in cellular therapy trials for regenerative medicine and immunomodulation [227,228]. Recently, MSCs have been shown to release exosomes, which may function as paracrine mediators between MSCs and target cells [229,230]. Furthermore, MSC-derived exosomes (MSC-Exos) have been found to recapitulate the biological activity of MSCs; as such, they may serve as an alternative therapy to MSCs [231]. Reports indicate that MSC-derived exosomes are ideal vehicles for the transportation and delivery of molecules including therapeutic genes, drugs, and RNA to targeted cells [232].

In the last decade, the use of exosomes in drug delivery has been considerably studied in AKI and has proven to be a promising strategy for the treatment. In one study, MSCs engineered to overexpress miRNA-let7c selectively targeted damaged kidneys and this upregulated miR-let7c-MSC therapy, attenuated kidney injury, and significantly decreased fibrosis by downregulating fibrosis markers, including collagen IVα1, metalloproteinase-9, transforming growth factor (TGF)-β1, and TGF-β type 1 receptor (TGF-βR1), in UUO mouse kidneys compared with non-targeted control MSCs [233]. In another study of I/R-induced injury in AKI, exosomes derived from MSCs pre-conditioned with melatonin were found to provide the best protective effect against kidney injury compared to therapy by MSCs or exosomes derived from non-preconditioned MSCs. This was demonstrated by a decreased kidney injury, reduced blood levels of kidney damage markers, declined oxidative stress status, inhibition of inflammation, and improved regeneration [234]. In another study, exosomes derived from melatonin-treated healthy MSCs were used to determine their therapeutic potential in CKD [235]. Treatment with melatonin increased the expression of cellular prion protein through the upregulation of miR-4516 in MSC-Exos. This study suggested that the treatment of CKD-MSCs with melatonin exosomes might be a strategy for the development of autologous MSC-based therapeutics for patients with CKD [235]. In another study, dexamethasone and glucocorticoid receptors were delivered to the kidneys by macrophage-cell-derived microvesicles expressing integrin surface markers. This attenuated renal injury with an enhanced therapeutic efficacy against renal inflammation and fibrosis in murine models of LPS- or adriamycin-induced nephropathy [236]. Other than biological components such as miRNAs, biochemical materials engineered to function with exosomes have also been reported to promote therapeutic efficacy in AKI. In one study, hydrogels containing Arg-Gly-Asp (RGD) peptide were produced and were functionally determined to sustain the retention and stability of EVs and to increase EV affinity. MSC-derived extracellular vesicles–RGD hydrogels facilitated MSC-derived let-7a-5p-containing EVs, improving therapeutic efficacy in renal repair in AKI [237].

In spite of the essential role that exosomes play in kidney disease pathology, their effectiveness still needs to be investigated. In addition, due to the low yield of exosomes, the research field of engineered exosomes is starting to be concentrated on the function of cell-derived nanovesicles [217,238]. So far, these studies demonstrate that engineered exosomes show beneficial effects on kidney injury. Furthermore, these studies have determined that loading other protective molecules in combination with biological materials or particles enhances protective effects toward kidney diseases and is worth further studies.

### 5.3. Exosome-Mediated Immunomodulation in Kidney Diseases

In addition to their aforementioned role as biomarkers and in drug delivery, exosomes are capable of influencing multiple aspects of the immune system. A number of studies have reported the vital role of exosomes in immunomodulation and immunosuppression. Amongst these studies, the role of exosomes in the regulation of inflammation in kidney injury has been extensively studied. Previous studies have pointed out that macrophages play important roles in the progress of AKI and AKI to CKD transition [239]. Macrophages adopt two distinct polarization states, switch between pro-inflammatory macrophages (M1s) or anti-inflammatory macrophages (M2s) activation phenotypes in response to varied external stimuli, and make different contributions to disease at different stages [240]. Exosomes derived from MSCs, multipotent cells with beneficial roles in tissue repair, have been found to contribute to immunomodulatory effects by regulating inflammation. For example, bone marrow MSC-derived exosomes (MSC-Exo) were reported to accelerate renal self-repair in ischemia–reperfusion injury (IRI)-induced mice by reducing M1 macrophage polarization and activating M2 macrophages polarization, thereby decreasing the levels of pro-inflammatory cytokines IL-1*β*, IL-6, and TNF-*α* while increasing the anti-inflammation factor IL-10. These anti-inflammatory effects were further amplified by indoleamine 2,3-dioxygenase overexpression in MSCs [241]. In addition to their role in immunomodulation, MSC-Exo has been reported to have a renoprotective function by regulating processes including cell proliferation, fibrosis, oxidative stress, autophagy, apoptosis, and necrosis (Table 5).

**Table 5 genes-14-01367-t005:** The mechanisms of MSC-derived exosomes in alleviating kidney injury.

Source of Exosome	Biological Material	Kidney Disease	Conclusions	References
mBMMSCs	CCR2	I/R injury—AKI	Suppressed CCL2 activity, alleviated inflammation.	[242]
mADSCs	miR-486	Diabetic nephropathy	Inhibited Smad1/mTOR signaling pathway, increased autophagy, and reduced podocyte apoptosis.	[243]
hWJMSCs	miR-15a/b, miR-16	I/R injury—AKI	Decreased CX3CL1 expression, alleviated inflammation.	[244]
	miR-30	I/R injury—AKI	Suppressed DRP1 and mitochondrial fragmentation, showed anti-apoptotic effects.	[245]
hP-MSCs	miR-200a-3p	I/R injury—AKI	Activated the Keap1-Nrf2 signaling pathway and exerted antioxidant effects.	[246]
hUSCs	miR-146a-5p	I/R injury—AKI	Degraded IRAK1 and inhibited NF-κB signaling pathway.	[247]
	miR-216a-5p	I/R injury—AKI	Downregulated PTEN, anti-apoptotic effect on HK-2 cells	[248]

Abbreviations used: mBMMSCs—mouse bone marrow mesenchymal stem cells; CCR2—C-C motif chemokine receptors 2; CCL2—C-C motif chemokine ligand 2; mADSCs—mouse adipose-derived stem cells; CX3CL1—C-X3-C motif chemokine ligand 1; DRP1—dynamin-related protein 1; hWJMSCs—human Wharton jelly mesenchymal stromal cells; hP-MSCs—human placenta-derived mesenchymal stem cells; hUSCs—human urine-derived stem cells; Keap1—Kelch-like ECH-associated protein 1; Nrf2—nuclear factor erythroid 2-related factor 2; IRAK1—degraded targeted and degraded the 3′UTR of interleukin-1 receptor-associated kinase 1 mRNA; PTEN—phosphatase and tensin homolog; HK-2—human proximal tubular epithelial cells.

Several studies have demonstrated that miRNAs contained within MSC-Exo play a role in immunosuppression. In one study, it was demonstrated that let-7b, a member of the let-7 miR family, contained in MSC-Exo induced the generation of the M2 immunosuppressive phenotype in renal macrophages. In addition, mice that received let-7b containing MSC-Exo showed lower concentrations of M1-derived pro-inflammatory cytokines TNF-α and IL-1β measured in I/R-injured kidneys [249]. In another study, it was demonstrated that miR-21 had anti-inflammatory properties and contributed to alleviating I/R-induced AKI [250]. In this study, the authors reported that miR-21 contained in MSC-Exo reduced NF-κB activity in the renal infiltration and maturation of dendritic cells (DCs). The administration of miR-21 containing MSC-Exo attenuated the production of pro-inflammatory cytokines such as IL-12, IL-6, and TNF-α, and reduced the activation of Th1 and Th17 cell-driven inflammation in I/R-injured kidneys [250].

In addition to MSC-Exo, miRNA-containing exosomes derived from renal TECs have been found to play a significant role in modulating inflammation. In a recent study, exosomal miR-19b-3p released from renal TECs was demonstrated to mediate the cross-talk between renal TECs and macrophages, which subsequently induced tubulointerstitial inflammation [251]. The authors found that miR-19b-3p was notably increased in TEC-derived exosomes from an LPS-induced AKI mouse model and in an adriamycin-induced chronic proteinuric kidney disease mouse model. They further reported that TEC-derived exosomal miR-19b-3p was internalized by macrophages, leading to M1 macrophage activation through the NF-κB/SOCS-1 pathway, thereby promoting tubulointerstitial inflammation. The administration of TEC-derived exosomes with miR-19b-3p inhibition significantly decreased the expression levels of IL-6, MCP-1, and iNOS mRNA in recipient macrophages and resulted in decreased renal inflammation [251].

Apart from the roles of MSC-Exos and TEC-derived exosomes on immunomodulation and immunosuppression in kidney injury, reports have shown that tumor-derived exosomes play a significant role in tumor progression by mediating the cross-talk between tumor cells and immune cells such as macrophages. In a recent study, it was demonstrated that RCC-derived exosomes containing a high amount of a new long non-coding RNA (lncRNA), named lncRNA activated in RCC with sunitinib resistance (lncARSR), played an important role in modulating inflammation [252]. LncARSR was found to interact with miR-34/miR- 449 to activate the STAT3 pathway, resulting in the polarization of the M1 to M2 macrophage phenotype in RCC cells. This polarization resulted in the secretion of anti-inflammatory factors such as IL-10, creating a more suitable microenvironment for tumor metastasis [252]. Sunitinib is a first-line targeted drug for metastatic RCC; however, its resistance is a major challenge for RCC patients. LncARSR correlated with a clinically poor sunitinib response and was found to promote sunitinib resistance by binding to miR-34/miR-449 and modulating the expression of AXL and c-MET in RCC cells [253]. Furthermore, resistant cells could secrete lncARSR via exosomes, transforming sunitinib-sensitive cells into resistant cells, thereby disseminating drug resistance. Targeting lncARSR was found to restore the sunitinib response, suggesting that lncARSR may serve as a predictor and a potential therapeutic target for sunitinib resistance in RCC [253].

Increasing evidence has demonstrated that exosome-containing miRNA derived from MSCs and TECs play a pivotal role in the immunomodulation and immunosuppression of kidney diseases, such as AKI and RCC. In addition, among the mechanisms by which these exosome-derived miRNA function, there is considerable evidence demonstrating that exosome-derived miRNAs function through macrophage polarization from M1 to M2 phenotypes. Taken together, these studies suggest that genetically modifying or engineering MSCs and TECs for the targeted delivery of miRNA may pave the way for an effective approach to improving kidney disease therapy.

## 6. Summary and Future Perspectives

Liquid biopsy has opened up a new avenue for the early diagnosis and treatment of kidney diseases, especially when tissue samples are difficult to or cannot be obtained. Using liquid biopsy and circulating biomarkers in kidney diseases allows for the assessment of molecular profiles of specific nephron cell. These circulating biomarkers could be used as non-invasive tools to be able to evaluate disease progression and the response to on-going treatments, thereby guiding the medical choice for further treatment strategies. However, there is the need for advanced molecular detection methods to enhance the functionality and reliability of liquid biopsy for clinical practice.

The potential of exosomes to reflect changes in the different segments of the nephron and their release in the urine emphasizes their role in the pathophysiology of kidney diseases. In the field of kidney biology and disease, the therapeutic applications of exosomes thus far comprise their use as prognostic and diagnostic biomarkers, drug delivery carriers, and cell-free therapy. Studies have particularly focused on the potential of exosomes as biomarkers, targeting primarily urinary exosomes, given the ease of collection and ability for urine to serve as a liquid biopsy. Several promising exosome-derived biomarkers have demonstrated the ability to detect early kidney damage and localized kidney injury and to predict disease progression and severity. Increasing evidence has also demonstrated the extraordinary therapeutic potential of MSCs in various kidney diseases, including ischemic diseases such as AKI. Currently, exosomes derived from various sources, including MSCs and TECs, have been reported to modulate several processes, including inflammation, fibrosis, and oxidative stress, which subsequently alleviates renal I/R injury and exerts reparative effects on the kidney. This suggests that MSC-derived and TEC-derived exosomes may serve as promising therapeutic candidates for ischemic AKI. Together, these studies demonstrate the potential of exosomes to improve the diagnostic and clinical treatment of kidney diseases.

Despite the advances outlined in this article, it should be noted that the translation of exosomes into clinical practice for kidney diseases is lacking. In a large number of preclinical studies, exosome-based therapeutics have shown significant efficacy in various kidney diseases. However, their clinical efficacy has not been up to par, with only a couple of clinical trials aiming to explore the clinical efficacy of exosomes against RCC. This suggests that there are still many challenges ahead and questions that need to be addressed. Some of the challenges that need to be resolved are the lack of standardization and consistency in exosome detection, isolation, and purification methods, low exosome yield, which cell type to use for exosome derivation, and the low stability and retention of MSC-derived exosomes. As it pertains to the exosome yield, previous studies have reported that the large-scale manufacturing of exosomes can be obtained via rapid purification. However, the viability of this technique still needs to be assessed with different cell types. Furthermore, the concept of utilizing exosomes as a drug delivery vehicle is a fast-growing area in kidney research. While drug-loaded exosomes may serve as a next-generation drug delivery system, they lack certain features to qualify as vehicles for drug delivery, such as their loading capacity. Exosomes need to be able to hold a considerable quantity of therapeutic cargoes to be able to deliver desirable therapeutic effects. However, since they already contain cargo, their loading capacity is low.

Aside from their inadequacy in their drug loading capacity, the targeting of therapeutic exosomes to specific kidney cells is essential. Fortunately, advanced biotechnological studies in exosome engineering with the goal of designing highly specialized exosomes capable of site-specific targeting to particular tissue or cell types and loading appropriate bioactive cargo into the lumen or the surface of the exosome is ongoing. As research for the development of more viable exosomes for therapeutic purposes in kidney diseases is ongoing, it may be helpful to take into account fundamental questions that remain unanswered. What are the physiological and pathological stimuli that lead to the production, release, and uptake of exosomes by different renal cells? What mechanisms determine the specificity of exosome targets? How does the kidney disease condition affect the composition and the levels of urinary exosomes? Addressing these questions may provide new mechanistic insights and set the stage for the sophisticated development and subsequent translation of exosomes as therapeutic intervention strategies in kidney diseases.

In conclusion, with the rising incidence of kidney diseases, the need for novel prognostic, diagnostic, and therapeutic strategies have never been greater. Studies up to date demonstrate that exosome-based clinical applications hold promise as a next-generation treatment trajectory for kidney diseases. However, the mass production of exosomes, selection of exosome donor cells, drug loading methods, and administration routes are all prominent issues that need to be resolved. With the continuous advancements in biotechnology, we expect these limitations to be overcome. In the end, exosome-based prognosis, diagnosis, and therapy have promising and important clinical significance in the field of nephrology.

## Figures and Tables

**Figure 1 genes-14-01367-f001:**
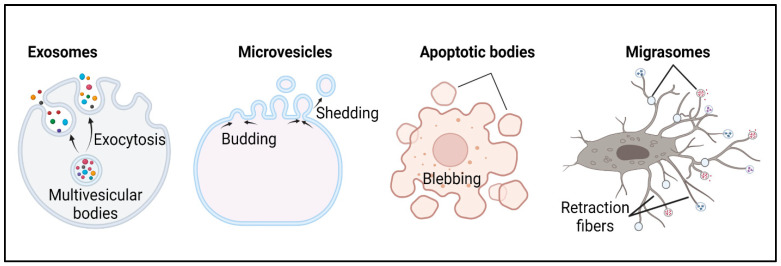
Extracellular vesicle classification. A depiction of the four different classes of extracellular vesicles indicating their modes of biogenesis and release. Exosomes are generated through the endocytosis of multivesicular bodies and are released via exocytosis, are spherical in shape, and vary in size. Microvesicles are generated and released through budding/shedding from plasma membrane, are irregular in shape, and vary in size. Apoptotic bodies are released through blebbing by cells undergoing apoptosis. Migrasomes (pomegranate-like structures) grow on and are released from the tips or intersections of retraction fibers, which mark the path of migrating cells.

**Figure 2 genes-14-01367-f002:**
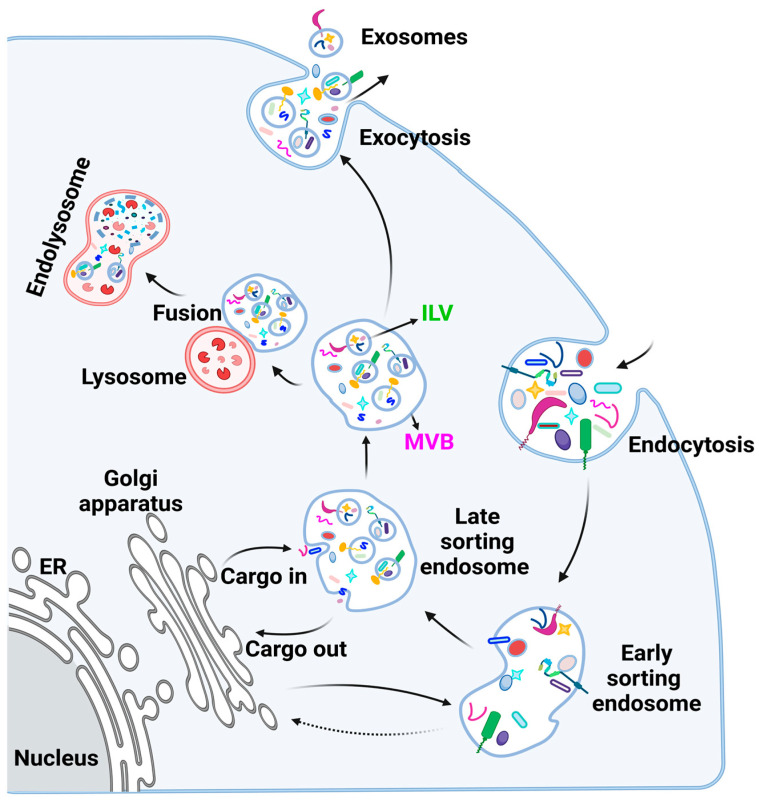
Exosome biogenesis. Exosomes are generated in a process that involves the double invagination of the plasma membrane in the early endosomes, resulting in the formation of intracellular multivesicular bodies (MVBs) that possess cell surface proteins and soluble proteins associated with the extracellular environment and the plasma membrane of the parent cell. Within these MVBs, inward invaginations occur, resulting in the formation of intraluminal vesicles (ILVs) in late endosomes following cargo sorting. Both ESCRT-dependent and ESCRT-independent driven pathways participate in creating multivesicular bodies. Exocytic MVBs fuse with the plasma membrane in Rab-GTPases-regulated manner. ILVs eventually become exosomes when secreted to the extracellular microenvironment. Exosome content depends on the cell type and the physiological and pathological condition of the cell.

**Figure 3 genes-14-01367-f003:**
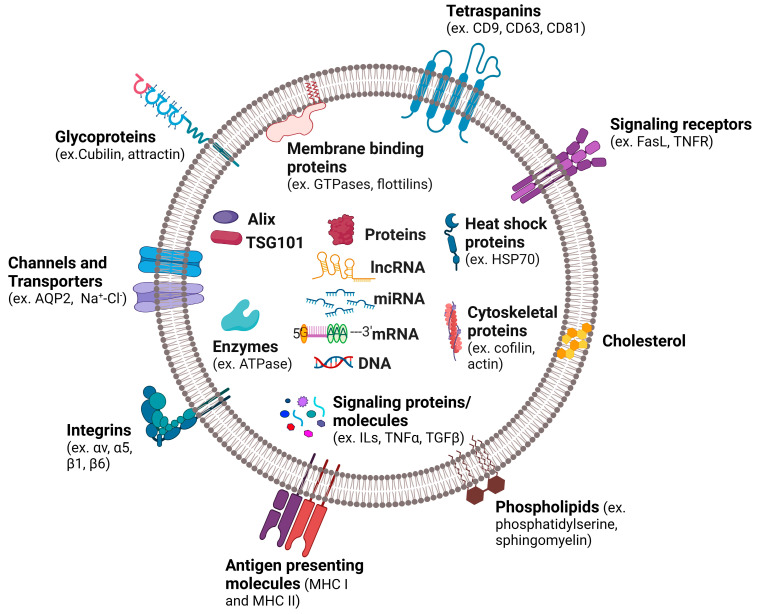
Hallmarks of exosomes. Exosomes contain a wide variety of molecules, such as nucleic acids, proteins, and lipids. The contents of the exosome are a representation of the cell of origin and of the physiological state of the cell from which the exosome is released from.

**Figure 4 genes-14-01367-f004:**
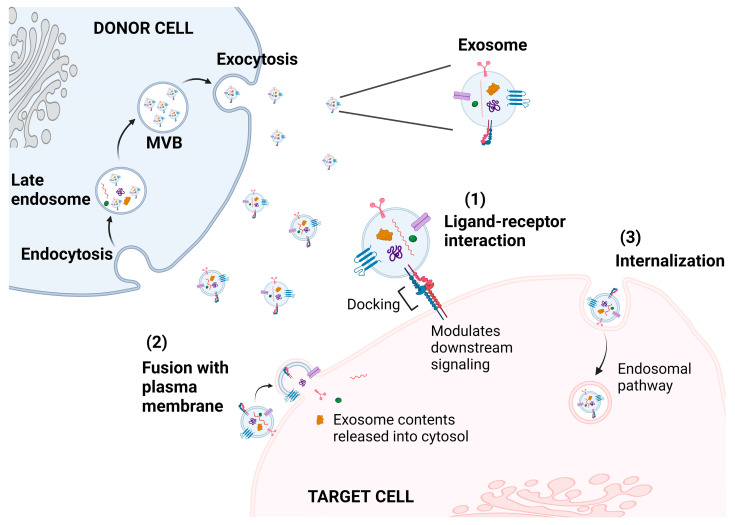
Exosome–recipient cell communication. Exosomes communicate with and modulate recipient cell signaling via multiple routes. Upon reaching the target cell, (1) ligands on the exosome surface membrane and receptors on the plasma membrane of the target cell can interact, inducing downstream signaling cascade in the recipient cell. (2) Exosome membrane can fuse with the plasma membrane and release its contents directly into the cytosol of the recipient cell. (3) Exosomes are internalized by the recipient cell and follow the endosomal pathway for cargo release/recycling.

**Table 1 genes-14-01367-t001:** Characterization of extracellular vesicles.

Feature	Exosomes	Microvesicles	Apoptotic Bodies	Migrasomes
Size	~30–150 nm	100–1000 nm	50–5000 nm	500–3000 nm
Morphology	Spherical or cup-shaped	Heterogenous	Heterogenous	Heterogenous
Biogenesis	Inward budding of MVBs formed in late endosome	Blebbing or outward budding of the plasma membrane	Repeated blebbing and retraction of plasma membrane of apoptotic cells	Grow on the tips of retraction fibers trailing behind migrating cells
Release	Fusion of MVBs with the plasma membrane (exocytosis)	Pinching/shedding of the plasma membrane	Blebbing of the plasma membrane	Breaking of the retraction fibers
Density	1.13–1.19 g/mL [51,52]	1.25–1.30 g/mL [53]	1.16–1.28 g/mL	N/A
Composition	Proteins, lipids, DNA, and different RNA species	Proteins, lipids, DNA, and different RNA species	Histones, cellular organelles, and fragmented DNA [54]	Contractile, cytoskeleton, cell adhesion and RNA-binding proteins, enzymes, and chaperons [37]
Markers	Tetraspanins (CD9, CD63, CD81), TSG101, Alix, flottilin-1 [35,55,56,57]	Integrins, selectins, CD40, flotillin-2	Annexin V (PS positive) [58]	TSPAN4, NDST1, PIGK, EOGT, CPQ [37,59,60]

Abbreviations used: MVB: multivesicular bodies, ALIX: apoptosis-linked gene-2-interacting protein X, TSG101: tumor susceptibility gene 101, TSPAN4: tetraspanin-4, PS: phosphatidylserine, NDST1: bifunctionalheparan sulfate N-deacetylase/N-sulfotransferase 1, PIGK: phosphatidylinositol glycan anchor biosynthesis class K, CPQ: carboxypeptidase Q and EOGT: EGF domain-specific O-linked N-acetylglucosaminetransferase.

**Table 2 genes-14-01367-t002:** Methods for exosome isolation and characterization.

Separation Technology	Specific Approach	Advantages	Limitations
Centrifugation-based technology	Ultracentrifugation [103,104,105]	Obtains highly purified exosome fractions	- Low yield, labor-intensive - Sediments other vesicles and protein aggregates - Good for exosomes purified from cell culture medium, but not bodily fluids
	Density gradient centrifugation [96,106]	Pure preparations	- Loss of sample - Time-consuming
Size-based technology	Size exclusion chromatography [105,107,108]	Preserves the integrity and activity of the exosome, no risk of vesicle aggregation and protein complex formation	- Run times are long - Breaks up larger vesicles, potentially skewing the results
	Ultrafiltration	Simple protocol, exosome yield has uniform size, high protein and RNA yield	- Low purity of exosomes
Exosome capture-based technology	Immune affinity capture/magnetic beads [35,109]	Highly specific and collects exosomes with high purity	- Low exosome yield - Not appropriate for purification of substantial amounts of exosomes
Polymer precipitation-based technology	Use of a solution containing polyethylene glycol, commercial kit most commonly used is ExoQuick [110,111,112,113,114,115]	Enriches exosomes from large volumes, less labor-intensive, permits reliable and high-throughput isolation of exosomes from low sample volumes	- Co-isolation of protein aggregates and non-vesicular contaminants such as lipo-proteins
Microfluidic-based technology	Immune-affinity, sieving and trapping on porous structures [116]	Extremely sensitive and quantitative analysis of exosomes	- Low exosome yield and presence of exosome aggregation

Therefore, the method of choice would depend on the sample source/type (cell culture media or bodily fluids) and the intended use for the exosomes. Moreover, the different isolation techniques can affect the analysis; as such, making a choice for the appropriate technique should be approached with caution [117].

**Table 3 genes-14-01367-t003:** Summary of the urine-derived exosomal biomarkers in kidney diseases.

Kidney Disease	Source of Exosome	Conclusions	References
AKI	Rat urine	Decreased exosomal AQP-1 in animals with renal IR.	[155]
	Human urine	Elevated exosomal Oat5 in cisplatin-induced AKI.	[156]
	Rat urine	Elevated exosomal Fetuin-A in cisplatin-induced AKI.	[157]
		Elevated exosomal Na^+^/H^+^ exchange type-3 in acute tubular necrosis.	[158]
	Human urine	Elevated exosomal NGAL and ATF3 in sepsis-induced AKI patients.	[157]
	Rat urine	Elevated exosomal levels of miR-16, miR-24, and miR-200c at an early phase of renal IR; elevated exosomal miR-125 and miR-351 at a late phase of renal IR.	[159]
CKD	Human urine	Decreased exosomal miR-29c associated with degree of renal fibrosis.	[160,161]
	Mouse kidney	Decreased exosomal miR-181a.	[162]
	Human urine	Decreased exosomal miR-200b was in CKD patients. The decrease was highest in exosomes derived from non-proximal tubule renal cells.	[163]
	Mouse kidney	Increased level of secreting transglutaminase-2 in UUO mice.	[164]
		Increased exosomal expression level of hsa_circ_0008925 in glomerular disease.	[165]
	Human urine	Increased exosomal expression level of has_circ_0036649 in glomerular disease.	[166]
	Human urine	Elevated exosomal bikunin precursor and histone-lysine N-methyltransferase but decreased VDAC1 in diabetic nephropathy patients.	[167]
	Human urine	Increased levels of AQP-2 and AQP-5 were detected in exosomes derived from diabetic nephropathy patients.	[168]
	Human urine	Increased levels of microRNAs such as miR-371b-5p, miR-320c, miR-572, miR-1234-5p, miR-6068, miR-6133, miR-4270, miR-4739, and miR-638 derived from exosomes in type 2 diabetic nephropathy patients.	[169]
	Human urine	Decreased levels of miR-30d-5p and miR-30e-5p in type 2 diabetic nephropathy patients.	[169]
	Human urine	Elevated levels of miR-15b, miR-30a, miR-34a, miR-133b, miR-342, and miR-636 in exosomes from type 2 diabetic nephropathy patients.	[170,171]
	Human urine	Elevated exosomal levels of let-7c-5p but decreased levels of miR-29c-5p and miR-15b-5p in type 2 DN patients.	[172]
PKD	Human urine	Decreased levels of PC-1 and PC-2 but increased level of TMEM2 in exosomes derived from ADPKD patients with *PKD1* mutation.	[15,173,174]
	Human urine	Increased expression of cystin and ADP-ribosylation factor-like 6 in PKD patients.	[173]
	Human urine	Increased levels of complement C3 and C9 in urinary EVs derived from ADPKD patients with or without progressive CKD; however, envoplakin, periplakin, and villin-1 levels were only increased in exosomes from ADPKD patients with progressive CKD.	[175]
	Human and rat urine	Increased exosomal level of AGS3 in PKD animals and patients.	[176]
	Human urine	Increased exosomal level of prominin 1 (CD133), cellular repressor of E1A-stimulated genes 1 (CREG1), and cadherin 4 in ADPKD patients.	[177]
RCC	Human urine	Increased levels of MMP-9, CP, PODXL, DKK4, and CAIX, and decreased levels of AQP1, CD10, DPEP 1, EMMPRIN, and syntenin-1 in the urinary exosomes of RCC patients.Increased levels of CP and PODXL could be used to distinguish RCC patients from healthy control individuals.	[178]
	Human urine	Increased level of miR-150-5p and decreased level of miR-126-3p have been reported. In addition, exosomal miR-126-3p combined with miR-449a could discriminate healthy individuals from ccRCC patients.	[179]
	Human urine	Exosomal miR-30c-5p and miR-204-5p could serve as potential diagnostic biomarkers for early-stage ccRCC	[180,181]

Abbreviations used: AQP—aquaporin; Oat5—organic anion transporter 5; Na^+^/H^+^—sodium/hydrogen; NGAL—neutrophil gelatinase-associated lipocalin; ATF3—activating transcription factor 3; UUO—unilateral ureteral obstruction; VDAC1—voltage-dependent anion-selective channel protein 1; DN—diabetic nephropathy; PC-1—polycystin-1; PC-2 —polycystin-2; TMEM2– transmembrane protein 2; AGS3—neutrophil gelatinase-associated lipocalin; CD133—prominin 1, CREG1—cellular repressor of E1A-stimulated genes 1; MMP-9—matrix metalloproteinase-9; CP—ceruloplasmin; PODXL—podocalyxin; DKK4—Dickkopf-related protein 4; CAIX—carbonic anhydrase IX; AQP1—aquaporin-1; CD10—neprilysin; DPEP 1—dipeptidase 1; EMMPRIN—extracellular matrix metalloproteinase inducer; ccRCC—clear cell renal cell carcinoma.

## Data Availability

Not applicable.

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
