# Peer review of "Liquid Biopsy at the Frontier of Kidney Diseases: Application of Exosomes in Diagnostics and Therapeutics"

_genes, 2023, doi:10.3390/genes14071367_

Round 1
Reviewer 1 Report
This review summarizes the application of exosomes in kidney disease. However, there are several concerns that need to be addressed:
1: The novelty of this manuscript should be clarified, considering the publication of numerous similar reviews on the subject. It is essential to highlight what distinguishes this review from others in terms of originality and unique perspectives.
2: In subheading 2, it is suggested to briefly address the Biogenesis, composition and characterization as these have been extensively covered in previous reviews. Emphasizing the advantages of exosomes in liquid biopsy would be more appropriate to support the focus of this review.
3: The title is about the exosomes from the liquid biopsy. But in part 5, the therapeutic application of exosomes focuses on those derived from MSCs. This inconsistency may confuse the audience. Moreover, the discussion of therapeutic applications in this review appears to be limited. Additionally, most of the cited references are over five years old, which raises concerns regarding the relevance and significance of publishing this review.
Author Response
1: The novelty of this manuscript should be clarified, considering the publication of numerous similar reviews on the subject. It is essential to highlight what distinguishes this review from others in terms of originality and unique perspectives.
Response: Thank you for the comment. We have modified the introduction to highlight the novelty of this manuscript. We pointed out areas in knowledge gap and provide recommendations.
2: In subheading 2, it is suggested to briefly address the Biogenesis, composition, and characterization as these have been extensively covered in previous reviews. Emphasizing the advantages of exosomes in liquid biopsy would be more appropriate to support the focus of this review.
Response: Thank you for the comment. We have included a paragraph in subheading 2, emphasizing on the advantages of exosomes in liquid biopsy. We also summarize the factors that can influence data collection with respect to urine sample handling and recommendations on how to improve pre-analytical handling.
3: The title is about the exosomes from the liquid biopsy. But in part 5, the therapeutic application of exosomes focuses on those derived from MSCs. This inconsistency may confuse the audience. Moreover, the discussion of therapeutic applications in this review appears to be limited. Additionally, most of the cited references are over five years old, which raises concerns regarding the relevance and significance of publishing this review.
Response: Thank you for the comment. The goal of this section was to emphasize on the therapeutic use of exosomes in immunomodulation. Our focus on MSCs particularly is based off reported literature. In the literature, majority of the studies report their findings in MSC-derived exosomes and in TEC-derived exosomes in kidney related diseases. As such, these were the systems we reported in the manuscript. Also, we have updated the references.
Reviewer 2 Report
This review article describes the novel diagnostic and therapeutic method for kidney diseases and the application of exosomes. The authors present the use of non-invasive tools, such as urine samples, to identify kidney disease. These contents have broad implications for publishing the present form. However, I have a few questions about the role of exosomes in kidney disease.
1. In clinical situations, kidney diseases are heterogeneous. Table 3 shows the urine-derived exosomal biomarkers, including AKI, CKD, PKD, and RCC. Is there additional information about glomerulonephritis, such as IgA nephropathy etc.?
2. Is there any difference between urine and blood exosomes as diagnostic biomarkers in kidney disease? The brief explanation might help to understand their roles in kidney diseases.
Author Response
1. In clinical situations, kidney diseases are heterogeneous. Table 3 shows the urine-derived exosomal biomarkers, including AKI, CKD, PKD, and RCC. Is there additional information about glomerulonephritis, such as IgA nephropathy etc.?
Response: Thank you for the comment. Yes, there is additional information about glomerulonephritis. However, we focused on AKI, CKD, PKD and RCC since they are areas of study interest in our research group.
2. Is there any difference between urine and blood exosomes as diagnostic biomarkers in kidney disease? The brief explanation might help to understand their roles in kidney diseases.
Response: Thank you for the comment. This concern has been addressed in subheading 2.6 in the manuscript.